# *Intersex* Plays a Role in Microbial Homeostasis in the Brown Planthopper

**DOI:** 10.3390/biology10090875

**Published:** 2021-09-06

**Authors:** Hou-Hong Zhang, Han-Jing Li, Yu-Xuan Ye, Ji-Chong Zhuo, Chuan-Xi Zhang

**Affiliations:** 1Institute of Insect Science, Zhejiang University, Hangzhou 310058, China; zhanghouhong@zju.edu.cn (H.-H.Z.); 21916101@zju.edu.cn (H.-J.L.); yeyuxuan@zju.edu.cn (Y.-X.Y.); 2State Key Laboratory for Managing Biotic and Chemical Threats to the Quality and Safety of Agro-Products, Key Laboratory of Biotechnology in Plant Protection of MOA of China and Zhejiang Province, Institute of Plant Virology, Ningbo University, Ningbo 315211, China; zhuojichong@nbu.edu.cn

**Keywords:** copulation, immune deference, microbiome, *Nilaparvata lugens*, RNA interference

## Abstract

**Simple Summary:**

RNAi-mediated knockdown of *intersex* in the newly emerged *Nilaparvata lugens* leads to abnormal expansion of the copulatory bursa by infection filled with bacteria. RNA-seq analysis shows a significant enrichment of immune defense genes responsive to bacteria in differentially expressed genes (DEGs). Moreover, inhibition of *intersex* expression by dsRNA treatment results in changes in the richness index of symbiotic microorganisms in copulatory bursa, fat body, and midgut of the planthopper. Specifically, significant changes are observed in the microbial community composition of the copulatory bursa. These findings reveal the function of *intersex* in maintaining microbial homeostasis in this insect, thereby providing insight to improve the pest control strategies.

**Abstract:**

Insects harbor a wide variety of symbiotic microorganisms that are capable of regulating host health and promoting host adaptation to their environment and food sources. However, there is little knowledge concerning the mechanisms that maintain the microbial community homeostasis within insects. In this study, we found that the *intersex* (*ix*) gene played an essential role in maintaining microbial homeostasis in the brown planthopper (BPH), *Nilaparvata lugens*. Injection of the double-strand RNA targeting *N. lugens ix* (*Nlix*) into the newly emerged females resulted in abnormal expansion of the copulatory bursa of BPH after mating. Further observation by transmission electron microscopy (TEM) revealed that the abnormally enlarged copulatory bursa resulting from ds*Nlix* treatment was full of microorganisms, while in contrast, the copulatory bursa of ds*GFP*-treated individuals stored a large number of sperm accompanied by a few bacteria. Moreover, RNA-seq analysis showed that the gene responses to bacteria were remarkably enriched in differentially expressed genes (DEGs). In addition, 16s rRNA sequencing indicated that, compared with control samples, changes in the composition of microbes presented in ds*Nlix*-treated copulatory bursa. Together, our results revealed the immune functions of the *Nlix* gene in maintaining microbial homeostasis and combating infection in BPH.

## 1. Introduction

Microorganisms, especially bacteria, are extensively distributed in the insect exoskeleton, gut, hemocoel, and even cells, which constitute an important part of insect body and have general impacts on many aspects of insect biology. Previous studies have demonstrated that certain resident microorganisms play vital roles in the biological process to promote insect fitness. For example, they can regulate insect development and reproduction [1,2], provide specific nutrients including essential amino acids or B vitamins [3,4,5,6], synthesize toxins or activate the insect immune system to protect their host insects against pathogens, parasitoids and other natural enemies [7,8,9,10], produce detoxifying toxins and resist against insecticides [11,12], and promote chemical communication among insects by specific components [13,14]. Furthermore, a variety and diversity of insect symbionts are related to the developmental stage, internal organs and tissues, and external environment factors. In *Bactrocera dorsalis* (Diptera: Tephritidae), the composition of microflora displays conspicuous differences among different developmental stages including larvae, pupae, and adults [15]. In *Apis mellifera* (Hymenoptera: Apidae), the age, caste, and seasonal variations determine the structure of symbiotic bacteria in the host [16]. Additionally, microbial components of *Laodelphax striatellus* (Hemiptera: Delphacidae) may be related to the longitude at which the host is located [17]. In general, community dynamics of endosymbiotic bacteria may facilitate insect adaptation.

To maintain host health, insects need to preserve the commensal microbial community homeostasis by resisting invasion from infectious microorganisms. It has been indicated that when infectious bacteria invade the intestine of *Drosophila melanogaster* (Diptera: Drosophilidae), the *duox* gene expresses a large number of reactive oxygen species (ROS) to combat pathogenic bacteria [18]. Inhibition of microbial infections by antimicrobial peptides modulated by the immune deficiency pathways has also been demonstrated [19]. Moreover, it is demonstrated that peptidoglycan recognition protein (PGRP) is involved in the innate immunity of *D. melanogaster* [20], and the maintenance of *B. dorsalis* microbiome at different developmental stages may be related to PGRP-encoding genes [21].

The brown planthopper (BPH), *Nilaparvata lugens* (Hemiptera: Delphacidae), a typical monophagous herbivore, is one of the most devastating rice pests in many Asian countries, which has caused serious losses to rice production every year [22]. Recently, the interaction between BPH and its symbionts has become the focus of attention in numerous researchers. The yeast-like endosymbiont (YLS), a symbiotic fungus present in a huge abdominal fat body cell [23], is capable of compensating BPH for amino acids that are lacking in the unbalanced nutrition diet sucking from the sap of the rice phloem [22,24,25]. *Arsenophonus* has been shown to generate an instrumental role in the provision of B vitamins to BPHs [22,26]. Furthermore, recent studies have shown that the bacterial communities are diverse and dynamic during BPH development [27], and microbial diversity differs between fat body and ovary [28].

The *intersex* (*ix*) gene is well conserved in insects, nevertheless, it has a multiplicity of functions in different species. For instance, it participates in sex differentiation, regulates external genital development, and female fecundity [29,30,31,32]. In our previous studies, we found that knocking out *Nlix* in nymphs led to the arrest of reproductive system development and failure of molting in BPHs. Meanwhile, the RNAi experiments performed on the newly emerged females led to abnormal enlargement of copulatory bursa, along with reduced fecundity and hatching rates [29]. Based on the phenotype of copulatory bursa after silencing *Nlix*, we hypothesized that the *ix* gene might be associated with the maintenance of the microbial community homeostasis in BPH. In this study, we further investigated the role of *Nlix* in microbial homeostasis at the time of copulation. Our results illustrated that the *Nlix* gene likely participated in the immune defense response of BPH to bacteria, and that RNAi-mediated knockdown of *Nlix* disrupted the immune system, consequently leading to the abnormal enlargement of the copulatory bursa due to infection in BPH.

## 2. Materials and Methods

### 2.1. Insect Rearing

In the field, BPH populations were originally obtained from rice crops in Hangzhou, China, in 2008. Insects were maintained on rice seedlings (variety xiushui 134) in a work-in chamber at Zhejiang University under the following conditions: 26 ± 1 °C, 60–70% relative humidity, and a 16 h light: 8 h dark photoperiod.

### 2.2. Expression Validation by Real-Time Quantitative PCR (RT-qPCR)

Total RNA was extracted from BPH using RNAiso Plus (Takara, Kyoto, Japan), and then 1 μg RNA was used to synthesize cDNA using the HiScript^®^ II Q RT SuperMix (Vazyme, Nanjing, China) through reverse transcription according to the manufacturer’s instructions. The target genes were quantified by RT-qPCR with an SYBR Color qPCR Master Mix kit (Vazyme, Nanjing, China) using a 20 μL reaction system consisting of 10 μL ChamQ SYBR Color qPCR Master Mix, 2 μL of 10-fold diluted cDNA, 0.6 μL of each primer, and 6.8 μL RNA-free deionized water. Specific primers for RT-qPCR designed by Primer Premier 6.0 software are listed in Table A1. The 18S rRNA gene of BPH (GenBank accession number JN662398.1) was used as the internal reference for RT-qPCR. The expression of target genes was validated by relative quantitative method (2^−ΔΔCt^). Each treatment was conducted in triplicate.

### 2.3. RNA Interference

The double-stranded RNAs (dsRNAs) of target genes were synthesized by using the T7 High Yield RNA Transcription Kit (Vazyme, Nanjing, China) from the amplified sequence. Two unique regions of *Nlix* were used as a template to synthesize dsRNA. The specific primers are presented in Table A1. ds*GFP* was utilized as control. Microinjection of BPH with dsRNA was carried out according to a previously described method [33]. Briefly, 100 ng of dsRNA was injected into the mesothorax of newly emerged females that had been anaesthetized with carbon dioxide for 10 s using a FemtoJet (Eppendorf-Netheler-Hinz, Hamburg, Germany). One hundred newly emerged females were used for dsRNA treatment and administered in three biological replicates. A set of 10 insects were collected as an independent sample to evaluate the RNAi efficiency of *Nlix* at three days after injection by RT-qPCR.

To determine the silencing effect of ds*Nlix* on BPH survival, newly emerged females were injected with dsRNA and reared on 10 cm high fresh rice seedlings for two days. Two groups, including one group treated with ds*Nlix* and the other one treated with ds*GFP* were selected to mate with the wild-type males, while the other two groups did not mate. Subsequently, all BPH individuals were reared for 6 days and survival rate of females was determined. ds*GFP* was injected as control. Each treatment was carried out in three biological replications.

### 2.4. Transmission Electron Microscope Analysis

We dissected the copulatory bursa from BPH females at three days after ds*Nlix* and ds*GFP* treatments for observation. In brief, newly emerged females were microinjected with dsRNA and kept under normal growth conditions for 2 days, mated for 1 day, and dissected to collect the copulatory bursa. Samples were prepared according to a previously reported method [34], and sections of processed samples were observed under a Hitachi Model H-7650 Transmission Electron Microscope (TEM).

### 2.5. Differential Expression Analysis Based on RNA-seq

Fifteen individuals, which were collected from the dsRNA-treated newly emerged females at 3 days after injection, were homogenized for total RNA extraction. The ds*GFP*- treated samples were used as control for nonspecific effects of dsRNA. Three sets of biological replicates were involved in each treatment. The cDNA library preparation and Illumina sequencing were performed by Novogene (Beijing, China). The clean reads were aligned with the reference genome by using HISAT2 [35]. The low-quality alignments were filtered with SAMtools [36]. TPM expression values were calculated by using feature counts for genes [37]. The DESeq2 package was utilized to detect differential expression analysis of the two groups [38]. The DEGs were selected upon the following thresholds: false discovery rate (FDR) *p* < 0.05 and absolute value of the log2 ratio >1. The raw data have been submitted to the National Center for Biotechnology Information Sequence Read Archive database under the accession number PRJNA755751.

### 2.6. Bacterial Community Characterization by Illumina Sequencing

The mated *N. lugens* female adults were selected for sample preparation at 3 days after injection. The surface of individuals was washed with 75% alcohol for 90 s and then rinsed thrice with sterile deionized water. Copulatory bursa, midgut, and fat body were then dissected using the sterile forceps in pre-chilled phosphorylation buffer (pH 7.4; 140 mmol/L NaCl, 2.7 mmol/L KCl, 10 mmol/L Na_2_HPO_4_, 1.8 mmol/L KH_2_PO_4_) under a dissecting microscope. Regarding 16S ribosomal RNA (rRNA) gene Illumina sequencing, three samples of copulatory bursa, midgut, and fat body were dissected and isolated from 60 insects, each with three biological replicates. All samples were immediately snap-frozen in liquid nitrogen and stored in a −80℃ refrigerator until DNA extraction.

Total DNA of each specimen was extracted using the QIAamp DNA Mini Kit (Qiagen, Hilden, Germany) following the manufacturer’s protocols. The quantity and quality of the extracted DNA were assessed using a NanoDrop NC2000 spectrophotometer (Thermo Fisher Scientific, Waltham, MA, USA) and agarose gel electrophoresis. The full-length bacterial 16S rRNA gene was amplified via PCR with the primer pair 27F (50-AGAGTTTGATCMTGGCTCAG-30) and 1492R (50-ACCTTGTTACGACTT-30). PCR products were purified with Agencourt AMPure XP Beads (Beckman Coulter, Indianapolis, IN) and quantified using the PicoGreen dsDNA Assay Kit (Invitrogen, Carlsbad, CA, USA). Subsequently, amplicons from each sample were pooled at equal concentrations and sent for sequencing using the Pacific Bioscience’s Sequel platform at Shanghai Personal Biotechnology Co., Ltd., in Shanghai, China. The raw data could be found in the National Center for Biotechnology Information Sequence Read Archive database under the accession number PRJNA755801.

The QIIME2 software was employed to explore the sequencing data as previously described. The amplicon sequence variants (ASVs) were de-replicated sequences generated after quality control using the DADA2 method, which were no longer clustered in similarity [39]. The representative sequences for each ASV were classified into organisms by a classify-sklearn algorithm using the Naïve Bayes classifier based on SILVA Database (https://www.arb-silva.de/; 7 January 2021) [40]. Besides, the alpha diversity indices were calculated, whereas the ASVs’ rarefaction and rank abundance curves were plotted by using QIIME2. Beta diversity analysis was performed using the Bray–Curtis distance index to confirm the structural variations in microbial communities of different samples, which were visualized by Principal Coordinate Analysis (PCoA) [41].

### 2.7. Statistical Analysis

The data are expressed as the mean ± SEM as shown in figures. GraphPad Prism 8.2.1 was used for statistical analysis. Difference between two groups was compared by using a two-tailed Student’s *t*-test (Figures 1B, 4A–F and 7E) and log-rank (Mantel–Cox) test (Figure 1A). The significance level was set at * *p* < 0.05, ** *p* < 0.01, and *** *p* < 0.001. The LEfSe method was used to analyze the differences in bacterial community composition between samples based on the one-against-all comparison strategy and the Wilcoxon test to determine biomarkers in different samples.

## 3. Results

### 3.1. Effects of Nlix Knockdown on the Survival and Copulatory Bursa Development of BPH Female Adults

An investigation was carried out to evaluate whether the abnormal enlargement of the copulatory bursa induced by ds*Nlix* treatment affected BPH survival. The ds*Nlix* treatment resulted in a high mortality in comparison with ds*GFP* treatment. In particular, compared with a survival rate of 74.0% for control groups, all the mated females died within 192 h after ds*Nlix* injection (x^2^ = 153.5, DF = 1, *p* < 0.001). Survival rate of unmated females treated with ds*Nlix* and ds*GFP* were 39.0% and 89.0%, respectively (x^2^ = 54.6, DF = 1, *p* < 0.001). Interestingly, we found that the survival rate of ds*Nlix*-treated females dropped dramatically at 24 h after mating, while that of the unmated ds*Nlix*-treated females decreased slowly at 108 h after injection (Figure 1A,B) (*t* = 102.0, DF = 4, *p* < 0.001). Furthermore, the size of copulatory bursa was not significantly different from control sample 72 h after *Nlix* knockout, which became enlarged quickly after mating; however, some of the unmated females had enlarged copulatory bursa at about 120 h after injection (Figure 1C–G).

### 3.2. Electron Microscopy Observations

As shown in Figure 2, the copulatory bursa of ds*GFP*-treated BPH was full of sperm, accompanied by a small number of bacteria. In contrast, injection of dsRNA for *Nlix* in the newly emerged females led to massive bacterial infections in the copulatory bursa, thereby preventing sperm from entering or surviving in copulatory bursa. TEM observation showed that the copulatory bursa might be abnormally enlarged due to bacterial infection after ds*Nlix* treatment, which in turn contributed to the death of BPHs.

### 3.3. Expression of Defense Response Genes Regulated by dsNlix Treatment

RNA-seq analysis was used to reveal genes potentially targeted by *Nlix*. A total of 183 DEGs were identified using transcriptome sequencing, among which, 130 including *Nlix* were significantly down-regulated and 53 were remarkably up-regulated after knockdown of *Nlix* in the newly emerged female adults. Gene Ontology (GO) enrichment analysis was performed to infer the biological processes of DEGs. It was found that the DEGs were mostly implicated in cuticle development (GO:0042335), chitin-based cuticle development (GO:0040003), and response to bacterium (GO:0009617) (Figure 3), confirming our assumption that the expression levels of immune defense response genes were altered in BPHs after treatment with ds*Nlix*. Results of qPCR assays demonstrated that 4 down-regulated genes and 1 up-regulated gene were associated with the response to bacterium, including ubiquitin carboxyl-terminal hydrolase (*scaffold.1090*, *t* = 5.926, DF = 4, *p* < 0.01), modular serine protease (*Nl.chr05.0253*, *t* = 11.06, DF = 4, *p* < 0.001) as well as genes involved in the melanin biosynthetic process from tyrosine (*Nl.chr11.425*, *t* = 24.69, DF = 4, *p* < 0.001, and *Nl.chr01.0531*, *t* = 17.86, DF = 4, *p* < 0.001), and in the vitamin catabolic processes (*Nl.chr06.0989*, *t* = 25.11, DF = 4, *p* < 0.001) as shown in Figure 4A.

As demonstrated by the qRT-PCR results, the expression level of each gene under RNAi treatment, in contrast to the control group, was significantly (*p* < 0.001) reduced (Figure 4C–F). The results suggested that there were no prominent differences in copulatory bursa between RNAi treatment and ds*GFP* treatment (Figure 5A–E). Nevertheless, knockdown of each of the three genes, *Nl.chr11.425*, *Nl.chr01.0531* or *Nl.chr06.0989*, by RNAi resulted in hypoplastic ovaries, and less oocytes in the ovaries (Figure 5B–D). Moreover, oviposition experiments showed that depletion of these three genes led to a significant reduction in the fecundity of BPH females, as shown in Figure 4B.

### 3.4. Microbial Community Compositions in Copulatory Bursa, Fat Body, and Midgut

The Shannon rarefaction curves based on 16S sequencing results for all samples almost reached the saturation plateau (Figure A1), indicating that the database produced from our samples was sufficient to capture the majority of microbial community information. At the phylum level, analysis confirmed that Proteobacteria and Firmicutes were the major microbial components in copulatory bursa of *N. lugens*. In fat body and midgut, Proteobacteria was the most predominant microbiota, which accounted for over 99% of the total phyla as show in Figure 6A. Among the bacterial genera, *Klebsiella*, *Lactococcus*, and *Enterobacter* were the major microbial members in copulatory bursa. *Arsenophonus* was the most predominant genus detected in fat body and midgut as shown in Figure 6B. Notable bacteria in the copulatory bursa were *Klebsiella*, *Lactococcus*, and *Enterobacter*, while *Arsenophonus* was the maker of both fat body and midgut (linear discriminant analysis (LDA) scores >4.5) as shown in Figure 6D. PCoA showed that microbiome in copulatory bursa was clustered separately from fat body and midgut based on the first principal component, in contrast, the samples from fat body and midgut tended to cluster together as shown in Figure 6C, suggesting a separation of the samples from copulatory bursa and the other two tissues/organs.

### 3.5. The Nlix Gene Was Associated with Shifted Microbial Communities of BPH

The changes in microbiota were evaluated after the depletion of *Nlix* gene in adults via dsRNA microinjection on the newly emerged females. We found that the expression level of *Nlix* gene was reduced at 72 h after dsRNA treatment (Figure 1B).

The Shannon rarefaction curves almost approached the saturation plateau, as shown in Figure A1, suggesting that the current sequencing database well represented the microbial communities in each library. In copulatory bursa, sample analysis revealed that the species of dominant microbial communities did not change, but the abundances of corresponding communities changed dynamically after ds*Nlix* treatment as shown in Figure 7A and Figure A2. Microbial groups with relative abundance exceeding 0.1% were selected for comparative analysis to avoid significant bias. As shown in Figure 7E, ds*Nlix* treatment increased the relative abundances of *candidatus saccharibacteria*, *Arsenophonus* and, *Ralstonia*, but decreased those of *Enterococcus* and *Acinetobacter*, compared with the ds*GFP*-treated samples. Besides, LEfSe analysis was employed to explain the difference between groups. The results showed that there were significant differences in the abundances of *Ralstonia*, *candidatus saccharibacteria*, *Enterococcus*, and *Acinetobacter* between RNAi and control groups as shown in Figure 7D. PCoA also revealed separation between ds*Nlix*-treated and ds*GFP*-treated groups. With regard to fat body and midgut, in contrast to copulatory bursa, there were no significant differences in microbial structure which was observed between ds*Nlix*-treated and ds*GFP*-treated samples as shown in Figure 7A,B. Besides, the number of ASVs in copulatory bursa, fat body, and midgut altered after ds*Nlix* treatment compared with ds*GFP* treatment; in other words, the microbial richness index changed as shown in Figure 7C.

## 4. Discussion

Insects are colonized by a vast array of symbiotic microorganisms, which may be one of the key factors in the successful adaptation of insects to their environment and food source [42]. In our study, we characterized the microbial community compositions of copulatory bursa, fat body, and midgut in BPH. Our results revealed that the genus of *Arsenophonus* was the most abundant in fat body, consistent with previous studies [28]. Simultaneously, microbial composition of the copulatory bursa as a part of the ovary were slightly different from the results of previous ovarian studies [28]. These may be attributed to the different external environments in which the hosts live, which supports the previous conclusion that the environment can greatly shape the microbial community structure [16,43]. Additionally, the genus of *Arsenophonus* was the most abundant in both the fat body and midgut, which might be attributed to its function of providing B vitamins to the host [22,26].

In our present study, we found that the copulatory bursa of control BPH individuals was teeming with a large number of sperm and a few bacteria; by contrast, the abnormally enlarged copulatory bursa caused by ds*Nlix* treatment was infected with a large variety of bacteria with no sperm, which was responsible for the dramatic reduction in the fecundity of BPH in our previous study [29]. Concurrently, a low survival rate was observed after 192 h of dsRNA injection in the newly emerged female adults, indicating that the *Nlix* gene played an essential role in maintaining the normal biological activity of BPHs. Furthermore, the mated females exhibited a lower survival rate after ds*Nlix* treatment than that of the unmated females, and the survival rate of females decreased sharply after 24 h of mating. This was probably owing to the fact that the mating behavior introduced bacteria that were not originally present in the BPH, thus accelerating the pathogenic infection. A study on the symbiotic community structure in the different developmental stages of BPH also demonstrates the differences in microbial community composition between male and female individuals [27]. These results prompted us to speculate that the *Nlix* gene was engaged in the immune defense response of BPH against bacterial infestation, and that the bacteria increased in copulatory bursa due to mating behavior may come from the environment as well as from the male, as copulation opens the mating orifice and exposes the male external genitalia to the external environment, making it easier for microorganisms in the environment to invade copulatory bursa.

RNA sequencing technology has become a powerful tool to study transcriptome profiling across a wide range of applications [44]. In our research, to investigate the DEGs of *Nlix*, RNA-seq was used to profile the transcriptome of female adults after RNAi treatment. Our results revealed that immune response genes that were responsive to bacteria were significantly enriched in DEGs, suggesting that *Nlix* gene might be a part of the immune defense network protecting BPH individuals from pathogenic bacteria. Moreover, when dsRNA treatment, which targeted each of these four down-regulated genes responsive to bacteria, was administered to BPH females in the newly emerged stage, we did not observe a similar phenotype in the copulatory bursa after the knockout of *Nlix* gene. However, a reduction in female fertility was observed after knockdown of genes *Nl.chr11.425*, *Nl.chr01.0531*, and *Nl.chr06.0989* by RNAi, revealing the complexity and pleiotropy of the immune process in the organism. Additionally, most of the DEGs were related to cuticle development, consistent with our previous observations that BPH exhibited lethal phenotype that failed to shed the old cuticle, which resulted from the depletion of *Nlix* [29].

Interestingly, results of 16S sequencing analysis showed the changes in microbial community structure in BPHs after *ix* gene knockout compared with the natural status, which might account for the abnormal enlargement observed in copulatory bursa after *Nlix* knockdown. Previous studies have shown that some regulatory factors contribute to the maintenance of the immune status in the host, which in turn regulates the composition of the symbiotic community. In *Drosophila*, proper expression of *duox* gene allows the host to achieve gut-microbe homeostasis by effectively removing pathogenic microbes, while tolerating the commensal bacteria [18]. Two *PGRP* genes have been demonstrated to be instrumental in regulating the community structure of Gram-negative and Gram-positive bacteria in *B. dorsalis* [21]. Genes encoding AMPs in some insects are expressed at the basal levels under conventional microbial loads and are systematically up-regulated in response to pathogenic microbial infections [19,45]. In our study, we also identified a similar immunomodulatory factor-*Nlix* gene; when its expression level was knocked out, changes were observed in the abundances of symbiotic bacteria in copulatory bursa, fat body, and midgut of BPH individuals. In particular, in the copulatory bursa, ds*Nlix* treatment led to an increase in its general richness index and changes in the abundances of some microbial taxa. It revealed the importance of *Nlix* genes in combating infection to preserve the microbial community homeostasis. Overall, the *Nlix* gene may be associated with the immune response to microorganisms in *N. lugens*.

Extraordinary experimental efforts have been conducted to elucidate the role of *Nlix* in biological processes of insects, such as growth and development, sex differentiation, and reproduction [29,30,31,32]. The function of *Nlix* in regulating microbial community composition was first discovered. However, the underlying mechanisms of the *Nlix* gene in the resistance to infection and maintenance of endosymbiotic community homeostasis in BPH requires further study.

## 5. Conclusions

In summary, we performed RNAi experiment in newly emerged BPH, in which *dsNlix* microinjection led to abnormal expansion of the copulatory bursa by infection filled with bacteria, which resulted in a high mortality rate of females after mating. Analysis of RNA-seq reveals a significant enrichment of immune defense genes responsive to bacteria in DEGs. Moreover, changes in the richness index of symbiotic microorganisms in the copulatory bursa, fat body and midgut of BPH were observed after inhibition of *Nlix* expression. Notably, the microbial community composition of the copulatory bursa was significantly altered. Consequently, these findings reveal the function of *intersex* in maintaining endosymbiotic community homeostasis in this insect and thus provide insight to improve the pest control strategies.

## Figures and Tables

**Figure 1 biology-10-00875-f001:**
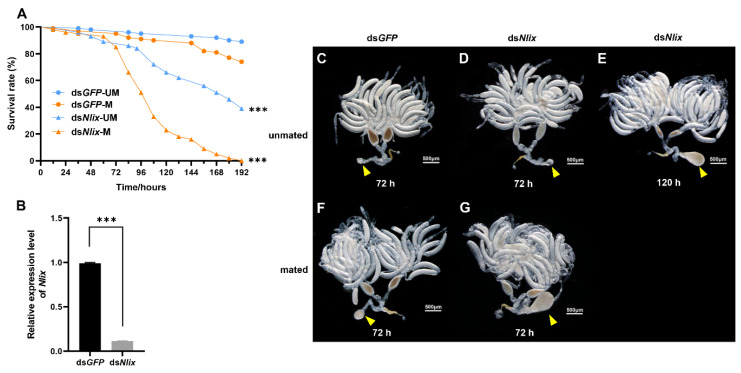
(**A**) Survival rate of BPHs at 192 h after RNAi when the newly emerged females were injected with dsRNA. ds*GFP* was used as control in each group, *n* = 100. Statistical analysis was performed by log-rank (Mantel–Cox) test. Survival rates were significantly decreased for both ds*Nlix*-UM compared to ds*GFP*-UM and ds*Nlix*-M compared to ds*GFP*-M. *** *p* < 0.001. M: samples mated with males; UM: unmated samples; (**B**) Transcript levels of *Nlix* after RNAi treatment in the newly emerged females. The RNAi efficiency of *Nlix* was evaluated by collecting 10 insects at 72 h after injection. Means ± SEM from three experiments. *** *p* < 0.001 (Student’s *t*-test); (**C**–**E**) Ovarian development phenotypes of unmated females after 72 h and 120 h of different treatments; (**F**,**G**) Ovarian development phenotypes of mated females after 72 h of different treatments. Yellow triangle: copulatory bursa.

**Figure 2 biology-10-00875-f002:**
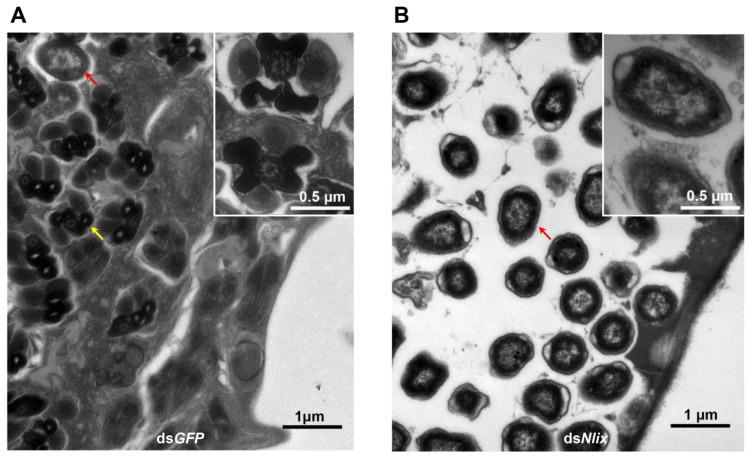
Transmission electron microscopy observation of copulatory bursa from the 72 h females in (**A**) the ds*GFP*-injected sample, (**B**) the ds*Nlix*-injected sample. A clearer view of the sperms/bacteria is shown in the top right corner of each panel. The yellow arrow marks the sperm and the red arrow marks the bacteria in the copulatory bursa.

**Figure 3 biology-10-00875-f003:**
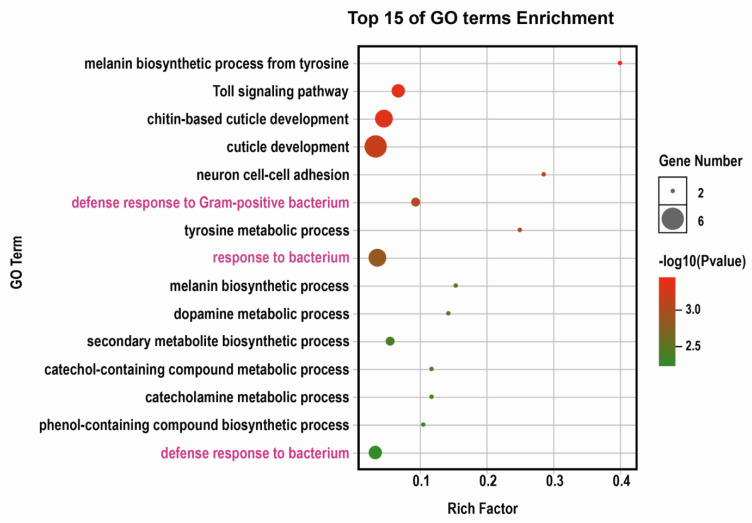
GO enrichment analysis of biological processes involved in the DEGs among female adults. GO enrichment study was performed using OmicShare tools. For data regarding GO terms, see Appendix A.

**Figure 4 biology-10-00875-f004:**
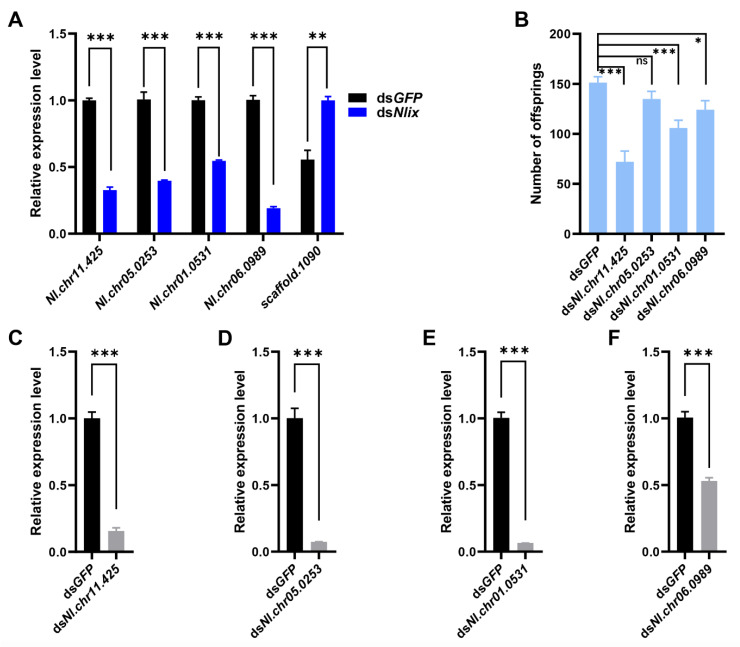
(**A**) qPCR verification of the 5 genes associated with the response to bacteria; (**B**) Number of the offspring produced from the treated females after knockdown of the 4 down-regulated genes associated with the response to bacteria, respectively. Each treatment was carried out in 10 biological replications; (**C**–**F**) Transcript levels of 4 down-regulated genes after RNAi treatment in the newly emerged females. At 72 h after injection, 10 insects were collected randomly to evaluate the RNAi efficiency. Means ± SEM from three experiments. * *p* < 0.05, ** *p* < 0.01, *** *p* < 0.001 (Student’s *t*-test); ns: no significant difference.

**Figure 5 biology-10-00875-f005:**
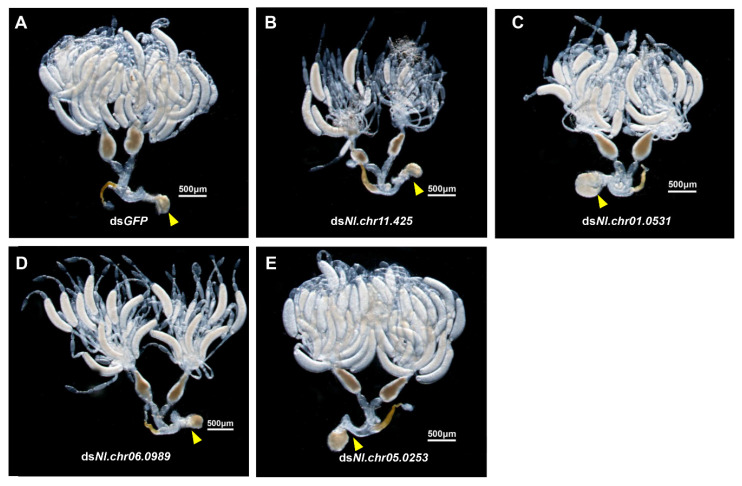
Phenotypes of ovary development after knockdown of 4 genes responsive to bacteria: (**A**–**E**) Ovaries dissected from the females treated with ds*GFP*, ds*Nl.chr11.425*, ds*Nl.chr01.0531*, ds*Nl.chr05.0253*, and ds*Nl.chr06.0989* for 72 h, respectively. Yellow triangle: copulatory bursa.

**Figure 6 biology-10-00875-f006:**
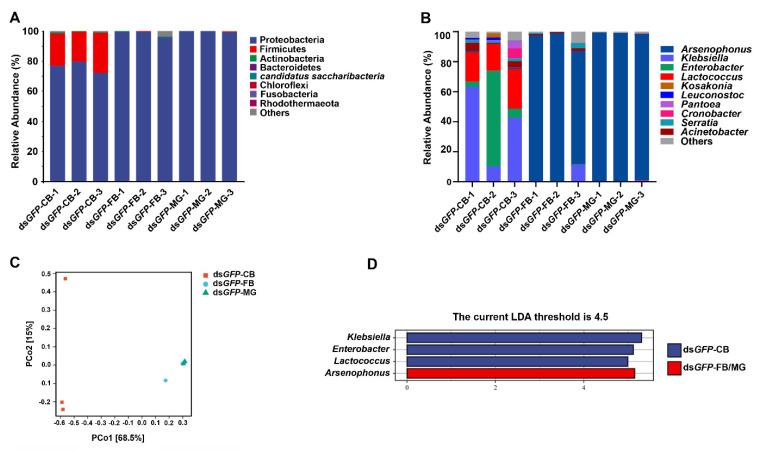
(**A**,**B**) Proportional compositions of microbiome in copulatory bursa (CB), fat body (FB), and midgut (MG) of *N. lugens* treated with ds*GFP*. Composition at the phylum (**A**) and genus (**B**) levels. The 10 species with the highest relative abundances were displayed, and the relative abundances of the remaining species were combined and grouped into Others; (**C**) Principal coordinates analysis (PCoA) plot of the Bray–Curtis distance based on the microbial structure in CB, FB, and MG; (**D**) LDA Effect Size (LEfSe) analysis based on ASV abundance in CB, FB, and MG. CB: copulatory bursa; FB: fat body; MG: midgut. For microbial taxon numbers, see Appendix A.

**Figure 7 biology-10-00875-f007:**
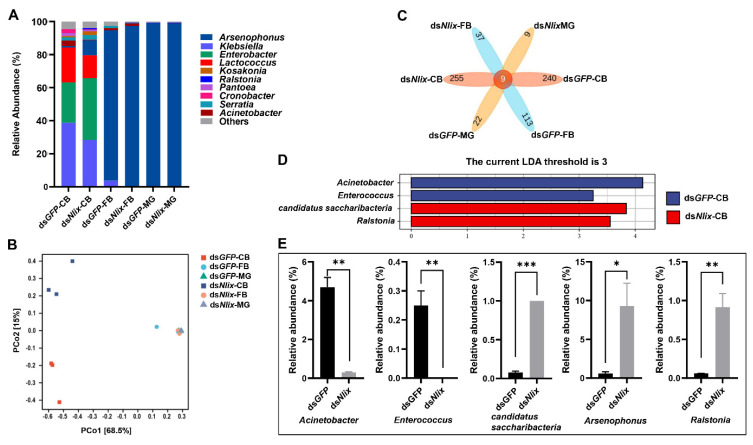
Microbial structural composition between ds*GFP*-treated group and ds*Nlix*-treated group: (**A**) Relative abundances of bacteria at the genus level in CB, FB, and MG of BPH in the ds*Nlix*-treated and control groups. The 10 species with the highest relative abundances were displayed, and the relative abundances of the remaining species were combined and grouped into Others; (**B**) Comparison of bacterial communities in samples treated with dsRNA via Bray–Curtis principal coordinate analysis; (**C**) Venn diagrams of the bacterial ASVs in CB, FB, and MG of BPH treated with dsRNA. The overlapping areas between blocks indicated the ASVs common to the corresponding groups, and the values in each block indicated the number of ASVs contained in the block; (**D**) LEfSe analysis based on ASV abundance in BC between ds*GFP*- and ds*Nlix*-treated samples; (**E**) Comparative analysis of abundance in the genera of *Acinetobacter*, *Enterococcus*, *candidate saccharibacteria*, *Arsenophous*, and *Ralstonia* under different treatments. *, **, and *** indicate significant differences between ds*GFP* treatment and ds*Nlix* treatment at *p* < 0.05, *p* < 0.01, and *p* < 0.001 levels, respectively. CB: copulatory bursa; FB: fat body; MG: midgut.

## Data Availability

The datasets generated for this study can be found in the the National Center for Biotechnology Information Sequence Read Archive database: PRJNA755751 and PRJNA755801.

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
