# Peer review of "Intersex* Plays a Role in Microbial Homeostasis in the Brown Planthopper"

_biology, 2021, doi:10.3390/biology10090875_

Round 1
Reviewer 1 Report
The manuscript gives a good contribution of Intersex plays a role in microbial homeostasis in the brown planthopper. However, I have concerns about the methods and results of the paper that I believe need to be addressed in order to improve its clarity. In particular, statistical methods (and data) are missing. Their approach is interesting but it has some flaws that make this version unacceptable for publication. Provided they conduct changes to the manuscript, I believe this paper could be of interest to the interested reader in insect-microorganism relationship. The annotated file is attached.

Author Response
Point to point response to reviewers
Reviewer 1
The manuscript gives a good contribution of Intersex plays a role in microbial homeostasis in the brown planthopper. However, I have concerns about the methods and results of the paper that I believe need to be addressed in order to improve its clarity. In particular, statistical methods (and data) are missing. Their approach is interesting but it has some flaws that make this version unacceptable for publication. Provided they conduct changes to the manuscript, I believe this paper could be of interest to the interested reader in insect-microorganism relationship. The annotated file is attached.
Response:
Thank you for the suggestion. We have added a “statistical analysis” paragraph in the materials and methods section in lines 191-197 as follows:
2.7. Statistical Analysis
The data are expressed as the mean ± SEM as shown in figures. GraphPad Prism 8.2.1 was used for statistical analysis. Difference between two groups was compared by using a two-tailed Student’s t-test and log-rank (Mantel-Cox) test. The significance level was set at * P < 0.05, ** P < 0.01 and *** P< 0.001. The LEfSe method was used to analyze the differences in bacterial community composition between samples based on the one-against-all comparison strategy and the Wilcoxon test, to determine biomarker in different samples.
In addition, data on GO terms and microbial taxonomic units can be viewed in the attachment. The raw data of RNA-seq and 16s rRNA sequencing have also been submitted to the National Center for Biotechnology Information Sequence Read Archive database under the accession number PRJNA755751.
We greatly appreciate the reviewer for his/her kind comments and revision suggestions in the PDF file. We revised one by one carefully following the reviewer’s kind suggestions. In addition, survival rate data were analyzed using the common log-rank (Mantel-Cox) test, as shown in Figure1.
Reviewer 2 Report
This is a nice paper with well designed and well-executed experiments to examine the role of the intersex gene in protecting planthoppers against pathogenic microorganisms at the time of copulation. The authors have used a wide array of detailed molecular and microscopic methods to track the changes in insect histology and bacterial composition following the knockdown of the intersex gene. Overall the results are clear and add to knowledge of the role of this gene - much of which comes from the authors' own research. The paper could be improved with a light revision of the English language and some reorganizing of text. For example, the authors already present their results at the end of the introduction - I think it is better to more clearly discuss the objectives and choice of methods here. On lines 181-183 - the authors have left the instructions for authors from the template. In lines 243-245 - the authors revert to describing methods - this should be in the methods section. In general the figure legends requires some further explanations, particularly figure 2 - for example, what are the arrows indicating? In terms of the methods, I do have one major concern, which I feel should be addressed - at least in the discussion. The authors have omitted one important control. As I understand it, the authors inject the insects with RNAi. This is not fully described in the methods and deserves further explanation. Injection presumably wounds the insect and opens the cuticle. It might then be expected that an insect with an injection wound would have bacterial infection. Can the authors sufficiently explain away this potential source of contamination. Furthermore, the authors should give some indication of where the 'pathogenic' bacteria have come from. If planthoppers were not interspersed after treatment - then it is possible that they picked up different contaminants from their cages, etc.; some further details on handling to avoid infections should be presented.
Author Response
Point to point response to reviewers
Reviewer 2
This is a nice paper with well designed and well-executed experiments to examine the role of the intersex gene in protecting planthoppers against pathogenic microorganisms at the time of copulation. The authors have used a wide array of detailed molecular and microscopic methods to track the changes in insect histology and bacterial composition following the knockdown of the intersex gene. Overall the results are clear and add to knowledge of the role of this gene - much of which comes from the authors' own research. The paper could be improved with a light revision of the English language and some reorganizing of text. For example, the authors already present their results at the end of the introduction - I think it is better to more clearly discuss the objectives and choice of methods here. On lines 181-183 - the authors have left the instructions for authors from the template. In lines 243-245 - the authors revert to describing methods - this should be in the methods section. In general the figure legends requires some further explanations, particularly figure 2 - for example, what are the arrows indicating? In terms of the methods, I do have one major concern, which I feel should be addressed - at least in the discussion. The authors have omitted one important control. As I understand it, the authors inject the insects with RNAi. This is not fully described in the methods and deserves further explanation. Injection presumably wounds the insect and opens the cuticle. It might then be expected that an insect with an injection wound would have bacterial infection. Can the authors sufficiently explain away this potential source of contamination. Furthermore, the authors should give some indication of where the 'pathogenic' bacteria have come from. If planthoppers were not interspersed after treatment - then it is possible that they picked up different contaminants from their cages, etc.; some further details on handling to avoid infections should be presented.
Comment: 1) the authors already present their results at the end of the introduction - I think it is better to more clearly discuss the objectives and choice of methods here
Response: Thanks. In lines 87-91, we added the objectives as follows: Based on the phenotype of copulatory bursa after silencing Nlix, we hypothesized that the ix gene might be associated with the maintenance of the microbial community homeostasis in BPH. In this study, we further investigated the role of Nlix in microbial homeostasis at the time of copulation. And we also added choice of methods in materials and methods section.
Comment: 2) On lines 181-183 - the authors have left the instructions for authors from the template
Response: Thanks. We have deleted the lines 181-183.
Comment: 3) In lines 243-245 - the authors revert to describing methods - this should be in the methods section.
Response: Thanks, we omitted lines 243-245 as this was described in materials and methods section.
Comment: 4) In general the figure legends requires some further explanations, particularly figure 2 - for example, what are the arrows indicating?
Response: Thanks. we have explained the legends in Figure 2 as follows: The yellow arrow marks the sperm and the red arrow marks the bacteria in the copulatory bursa.
Comment: 5) In terms of the methods, I do have one major concern, which I feel should be addressed - at least in the discussion.
Response: Thanks. We added the methods of statistical analysis in materials and methods section as mentioned earlier.
Comment: 6) The authors have omitted one important control
Response: We used dsRNA of the green fluorescent protein (dsGFP) of Aequorea victoria as a negative control. dsGFP is extensively used as RNAi control.
Comment: 7) the authors inject the insects with RNAi. This is not fully described in the methods and deserves further explanation.
Response: In lines 123-127, we added the injection method of RNAi as follows: emerged females that had been anaesthetized with carbon dioxide for 10 s using a FemtoJet (Eppendorf-Netheler-Hinz, Hamburg, Germany). one hundred newly emerged females were used for dsRNA treatment and administered in three biological replicates.
Comment: 8) Injection presumably wounds the insect and opens the cuticle. It might then be expected that an insect with an injection wound would have bacterial infection. Can the authors sufficiently explain away this potential source of contamination. Furthermore, the authors should give some indication of where the 'pathogenic' bacteria have come from. If planthoppers were not interspersed after treatment - then it is possible that they picked up different contaminants from their cages, etc.; some further details on handling to avoid infections should be presented.
Response: We appreciate the reviewer for his/her positive comments and suggestions. In discussion section, we added explanations about the potential resource of bacteria in lines 391-395 as follows: and that the bacteria increased in copulatory bursa due to mating behavior may come from the environment as well as from the male, as mating opens the mating orifice and exposes the male external genitalia to the external environment, making it easier for microorganisms in the environment to invade copulatory bursa.
Round 2
Reviewer 1 Report
The revised manuscript "Intersex plays a role in microbial homeostasis in the brown planthopper" can be accepted for publication. Just some typos should be corrected, as follows:
L.35: Delete “;” after RNA interference
L.54: Change “Homoptera” by “Hemiptera”
L.64: … of D. melanogaster [20], and…
L.87: … time of copulation. Our…
L.85: … (variety xiushui 134)…
Ls.106 and 115: Change “N. lugens” by “BPH”
L.146: Pull apart “(FDR) P < 0.05”. In p-values, letter P should be minuscule. Check in all manuscript
Ls.192-193:
L.193: Sentence starting “The dsNlix treatment resulted…”
Ls.196-200: For these results, provide statistical values for Mantel-Cox (X2 =?, DF =?, and p-value) and Student’s (t =?, DF =?, and p-value) test
Ls.226-239: Again, provide statistical values for each Student’s (t =?, DF =?, and p-value) test
L.284: …midgut. Microbial taxon…
L.321: …food source [42]. In…
L.324: …previous studies [28].
L.326: …ovarian studies [28].
L.331: …host [22,26].
L.345: … female individuals [27].
L.352: … of applications [44]. In
L.357: Delete “significantly”
L.365: …of Nlix [29].
L.373: …commensal bacteria [18].
L.377: …microbial infections [19,45].
L.387: … and reproduction [29-32].
L.411: …MG: midgut.
Ls.413-511: Check references according to the journal style.

Author Response
We greatly appreciated the reviewer 1 for his/her careful correction for our typos. All the reviewer’s comments have been revised as suggestions. Attached please find the revised version.